# Evaluating and Finetuning Models For Financial Time Series Forecasting

## Abstract

Time series forecasting is challenging as it is subject to a lot of noise, and the predictions often depend on external events. Still, recent deep learning techniques advanced the state-of-the-art on specific datasets, while they keep failing on other noisy datasets. This paper studies the case of financial time series forecasting, a problem that exhibits both a high noise and many unknown dependencies. We will show that the current evaluation pipelines are imperfect and forget a trivial baseline that can beat most models. We propose a new evaluation pipeline that is better suited for our task, and we run this pipeline on recent models. This pipeline is based on deciding which assets to buy and sell rather than predicting exact prices. Next, as the small datasets used in current approaches limit the models' size, we train a general model on a massive dataset (containing a hundred times more data points than existing datasets) and show this model can be finetuned to improve the performance on small datasets. All our code and models will be published to help the community bootstrap and evaluate their future models.

## 1 Introduction

Financial markets play a significant role in our modern world as they regulate the global economy and are directly interlinked with our daily life: Any important event (pandemic, war) disturb the stock prices, and, vice-versa, changes in stock prices also lead to observable real-life consequences (sub-prime crisis, inflation). Therefore, it is crucial for the public interest to understand and regulate how these markets work.

Yet, the financial world is secret: Almost no data is available to the general audience, and only big firms can get insights. This observation makes academic research (particularly in data science) extremely difficult and forces us to rely on small public datasets, limiting the models we can construct.

These datasets are generally used to predict the future in one way or another. Given a price history, we try to predict future prices (forecasting) or price directions (classification). These tasks raise many challenges typical of time series that are emphasized in the case of finance.

First, the amount of noise is significant, and understanding this noise is hard in itself (Black, 1986). This noise comes from many sources, mainly program trading (i.e., automatic trading) (Li et al., 2020). It makes predictions, notably short-term predictions, extremely hard as the signal is barely visible.

Second, real-life events, such as political news, financial reports, or scientific discoveries, impact price movements (Yang et al., 2015; Mo et al., 2016). This impact can also be observed with fake news (Kogan et al., 2019). Such exogenous variables make the prediction using solely price history extremely difficult, especially with few data. However, this is a common task tackled by state-of-the-art models.

Third, financial markets are highly dynamic systems with many actors: A new model with an edge (i.e., a way to outperform the market) will be slowly arbitraged away, i.e., it will lose its edge with time (Krauss et al., 2017). Many reasons explain this fading: The actors change with time, competitors adapt, new algorithms appear, models are reverse-engineered, etc. This means that having a model that performed well in the past will not necessarily have an advantage in the future. Besides, using old data for training can be useless as they come from a very different ecosystem.

Despite all these challenges and the area's competitiveness, we regularly observe that new papers claim to improve the state-of-the-art. In this article, we try to understand how it is possible with so few data and many unknowns. We will show that the evaluation pipeline is imperfect and forgets a trivial baseline that beats the most complex models on usual metrics. Then, we will propose a new evaluation pipeline better suited for forecasting financial time series.

Next, we will show that it is possible to build a large dataset using web scrapping tools. We will use this dataset to train and evaluate large models with our new pipeline. In addition, we will show that these models can be a general building block that can be finetuned to solve specific tasks on smaller datasets like it is done for natural language processing. These models will be publicly available.

Overall, our contributions are the following:

1. We introduce a new evaluation pipeline better suited for financial time series.
2. We compare state-of-the-art deep learning methods for financial time series forecasting.
3. We train large models on a large dataset.
4. We show that these models can be used to solve more specific tasks.

## 2 PREVIOUS WORK

**Datasets**    Most of the time, financial time series are mixed with other time series and evaluated together or separately. We found four primary datasets used to train publicly available models. Some models were trained on datasets we cannot access. For each dataset, we report in Table 1 the number of data points, the number of time series, the time span of the data, and the granularity (i.e., the interval between two consecutive data points). Note that the number of time series can be artificially high because longer series were cut into smaller ones.

The oldest dataset is M3 (Makridakis & Hibon, 2000) and was succeeded by M4 (Makridakis et al., 2020). These datasets contain a mix of time series, among which financial time series. On these datasets, the data can be very old and has no constant granularity, but it is a monthly granularity on average. The Exchange Rate dataset (Wu et al., 2021) contains eight time series representing the exchange rate between pairs of currencies. It is often used with other datasets (electricity, traffic, illness). The NASDAQ 100 dataset (Qin et al., 2017) contains companies' stock prices in this index over a short period. Some works used a similar dataset for S&P500 but without publishing the data, and therefore, no comparisons are possible with newer models. We reported these datasets in Table 1, in addition to our new dataset (see Section 4).

We will not directly use the datasets presented here as they do not contain timestamps and little or no alignment between the time series. However, these are crucial properties for time series forecasting as we cannot predict past values with training data from the future. Besides, an investment strategy decides what to buy or sell at a given time, given the current market's state (i.e., the value of all the assets simultaneously, see Section 4). We will extract similar datasets from our new dataset.

| Dataset | #points | #time series | time span | granularity |
|---|---|---|---|---|
| M3 (Makridakis & Hibon, 2000) | 28k | 308 | 1947-1989 | ∼month |
| M4 (Makridakis et al., 2020) | 8.3M | 25k | 1944-2019 | ∼month |
| Exchange Rate (Wu et al., 2021) | 60k | 8 | 1990-2016 | ∼daily |
| NASDAQ 100 (Qin et al., 2017) | 40k | 80 | 2016-2017 | ∼minute |
| Ours (uncut, uncleaned) | 12B | 118k | 2010-2023 | ∼minute |
| Ours (72) | 992M | 14M (22k*) | 2010-2023 | ∼minute |
| Ours (360) | 486M | 1.3M (15k*) | 2010-2023 | ∼minute |
| Ours-S&P500 (72) | 116M | 1.6M(500*) | 2010-2023 | ∼minute |
| Ours-CAC40 (72) | 1.7M | 24k (40*) | 2010-2023 | ∼minute |

Table 1: Comparison of the datasets in the state-of-the-art. *: We cut long time series.

**Financial Time Series Forecasting**    There are several financial time series forecasting types. First, most approaches differentiate short-term predictions (a few steps in the future) from long-term forecasts (many steps in the future). In the literature, we observed horizons (distance of the prediction)

ranging from one step to around 700 steps, which at the granularity of one hour is about two months. Some approaches consider univariate and multivariate forecasting, in which more than one time series is given as input (for example, to predict the price of an index (Qin et al., 2017)). Then, many models were proposed. Most are based on convolutional neural networks, recurrent neural networks, or Transformers (Vaswani et al., 2017). These standard networks are often used as baselines (Wang et al., 2022; Xu et al., 2023).

N-BEATS (Oreshkin et al., 2019) tries to decompose the input into several signals using specific blocks and residual connections. Later, N-HITS (Challu et al., 2022) improves this architecture by adding pooling and hierarchical interpolation to have a multi-scale approach. Autoformer (Wu et al., 2021) is a variation of Transformers that decomposes the input into a trend and a seasonal component. The traditional attention mechanism is replaced by an autocorrelation block that compares the time series with a shifted version of the time series. SciNet (Liu et al., 2021) also tries to have a multi-scale approach. It is a network that progressively downsamples the input and creates high-level features. DA-RNN (Qin et al., 2017) uses a combination of attention and Recurrent Neural Networks (RNN) to learn meaningful representations. DSTP-RNN (Lahmiri, 2016) is a variation on DA-RNN that separates spatial and temporal representations. DSANet (Huang et al., 2019) is another variation that uses convolutions and attention.

## 3 PROBLEM AND DEFINITIONS

Let $X = (X_1, X_2, ..., X_N)$ a time series of $N$ steps where $X_i \in \mathbb{R}^d$, with $d$ the dimension of each point in $X$. We are interested in forecasting a single point at a horizon $H$ given a context of size $C$, i.e., for $k < N - C - H$ and given $X_k, ..., X_{k+C}$, we want to predict $X_{k+C+H}$. In practice, we will focus on predicting a single dimension of $X_{k+C+H}$, the stock closing price.

Some previous works predict all the points $X_{k+C+1}, ..., X_{k+C+H}$ between the end of the context and the horizon. This task is more challenging because short-term forecasting is more subject to noise, whereas long-term forecasting might allow a trend to appear. The existence of these two problems created confusion in the literature. For example, SciNet (Liu et al., 2021) predicts a single point but copies the results from previous papers that predict multiple points (without rerunning the experiments), making the comparison irrelevant.

In practice, $X_i$ is almost always composed of a single value, the closing price at the given time $i$. In this paper, we also investigate the addition of open, low, and high prices on the interval $]i-1; i]$, i.e., the first price, the lowest price, and the highest price in the interval, as well as the volume (i.e., the number) of transactions on the interval.

As pointed out in the introduction, recent time periods are more complex to predict than older periods. Therefore, creating the training, validation, and testing data splits using the time components is essential. This was not always done in previous works (time and order are missing from M3 and M4, for example).

Finally, we introduce the notion of portfolio. A static portfolio $P$ is a pair of assets (represented as time series) $X^1, ..., X^K$ and weights $w_1, ..., w_K$ where each weight $i$ is the number of asset $i$ in the portfolio. We only consider positive weights in this paper. The value of a portfolio at time $t$ is: $P_t = \sum_{i=1}^{K}(x_k^i * w_i)$. Note that some portfolios are dynamic, i.e., their assets and weights might change through time.

## 4 METHODOLOGY

### 4.1 DATASET CONSTRUCTION

Our goal was to construct a large dataset of financial time series. To do so, we cleverly used web scrapping tools. Although this is not new in itself (Hajizadeh et al., 2012; Jagwani et al., 2018; Budiharto, 2021), we did it at scale and for a diversity of instruments (stocks, futures, indexes, ETF, options) that is missing in the literature. That allowed us to create a large and diverse dataset with a small granularity (5 minutes) and focus on recent time periods. The statistics of our dataset are presented in Table 1 in three versions: One with uncleaned and raw data, one where large time series are cut into smaller time series of 72 points (one day), and one where they are cut into 360 points

(one week). When we cut the dataset, we only keep the clean parts where the granularity is small enough, no data is missing, and there are no apparent outliers (return above a certain threshold). This explains the difference in size between the different versions.

To test our data on smaller and more used assets, we extract from our new dataset subsets for several indexes with the last known components in August 2023: S&P500 and CAC40. We get several markets (US and French) with different sizes (see Table 1).

## 4.2 Baselines

This paper will compare and retrain several baselines: DA-RNN, SciNet, Transformers, and a trivial baseline that always predicts the last known point or no price movement (we call it LAST). This baseline is surprisingly hard to beat and was totally forgotten in previous works. Therefore, we reran several experiments from the state-of-the-art using the code and data provided by the authors when available (to be sure we use the same evaluation setting). We were astonished that no model beat LAST, which shows that (1) the claimed improvements are just noise and (2) the traditional metrics used for the evaluation might not be adapted for financial time series forecasting. In this paper, we will rerun these baselines (for uniformity) and observe the same problems.

If we look in detail, the LAST baseline is, in fact, very bad from an investor point of view: It makes no decision to sell or buy assets (and therefore would perform poorly as a classifier) and makes no difference between all assets and all price histories. An investor would like to know which assets will perform best and worst to construct a profitable portfolio (see Section 4.5).

During the general pretraining, we want to predict $y_{t+1}$ given $y_0, ..., y_t$ for each baseline. For Transformers, we can predict $y_1, ..., y_{t+1}$ by carefully making the prediction solely using previous values. This has the advantage of producing more training data and having a flexible context size. Most models only take a predefined context length as input, whereas the context of Transformers is flexible. It can, therefore, adapt to new problems better.

## 4.3 Forecasting Metrics

For the evaluation, we will use the standard metrics for forecasting: the Root Mean Square Error (RMSE) and the Mean Absolute Percentage Error (MAPE). For a target variable $y = y_1, ..., y_n$ and a predicted value $\hat{y} = \hat{y_1}, ..., \hat{y_n}$, we have:

$$RMSE(y, \hat{y}) = \sqrt{\frac{\sum_{t=1}^{n}(y_t - \hat{y_t})^2}{n}} \quad \text{and} \quad MAPE(y, \hat{y}) = \frac{100}{n}\sum_{t=1}^{n}\left|\frac{y_t - \hat{y_t}}{y_t}\right|$$

## 4.4 Portfolio Evaluation Algorithm

This section presents our new evaluation algorithm to evaluate forecasting models for financial time series. This algorithm is inspired by the evaluation of financial strategies that must decide when to sell and buy instruments (Aragon et al., 2007; Strong, 2006). In this paper, we move beyond forecasting and focus more on determining the best choice (buy or sell) at a given time solely using a forecasting model. We aim to evaluate forecasting models that can later be used or finetuned by more advanced models, for example using reinforcement learning. Concretely, the new evaluation emphasizes **ranking**: We decide which assets are the best and the worst among a set of assets.

We proceed as follows. We split long time series (some contain 13 years of data) into smaller ones and align them in pools. A pool is a set of aligned assets that start and finish simultaneously. It represents a moment in time when we need to decide which assets to buy or sell. Then, we can compute the performance of our metric on each pool and take average metrics.

Our algorithm takes as input: Time series $X^0, ..., X^N$ containing $T$ points (also works if the time series are of different lengths as soon as we can construct pools), a context size $C$, a horizon $H$, a pool number $K$, and a forecasting model $M$. It is composed of the following steps:

1. Select the set of start indexes $S_{pool} = s_1, ..., s_K$ for the construction of the pools. This can be done randomly or with a rolling window strategy. In this paper, we take $S_{pool} = \{k * C \mid 0 \leq k < T/C\}$.

2. For each index $s_i$, create a pool $Pool_i$ of all the sub-assets starting at $s_i$ and finishing at $s_i + C$. Also, take the horizon point at $s_i + C + H$.

3. For each pool $Pool_i$, create a portfolio using the assets in the pool and the model $M$ (see Section 4.5). Then, compute the return of this portfolio after the horizon. Concretely, if we call $P_t$ the value of the portfolio at any time step (i.e., the weighted sum of the closing price of all the components), we compute $R_j^i = \frac{P_{s_i + C + H}}{P_{s_i + C}} - 1$.

4. We call $R = \{R_j^i\}$. We can compute the metrics:

   (a) Expected Return: $\mathbb{E}[R]$

   (b) Risk (standard deviation): $\sigma(R)$

   (c) Sharpe Ratio: $\frac{\mathbb{E}[R]}{\sigma(R)}$. We ignore the risk-free rate here as it is the same for all the models. This rate is generally equal to the rate of the American Treasury Bonds.

An extreme case is interesting to study. If we have $K = T$, we evaluate our model on all available points. This is interesting if we have a few training data points. However, with larger datasets, only evaluating a sample is often enough to get statistically significant results, and the computations are faster.

Later, we will present an annualized version of the expected return, risk, and Sharpe ratio to have more standard numbers. This annualization is done by multiplying by the number of trades in a year (or its square root). Besides, for the Sharpe ratio, we report the 95% confidence interval as given in (Lo, 2002).

## 4.5 FROM FORECASTING TO PORTFOLIO

We suppose we have a forecasting model $M_H$ for a horizon $H$. We aim to build a strategy using solely $M_H$, i.e., to construct a portfolio $P$ from a set of assets $X^1, ..., X^N$ (we do not need to use all of them).

Suppose we want to buy one unit of assets at a time $t$ (i.e., the sum of the weights is one; we suppose they are all positive here) and take our profit or loss after the horizon $H$ at $t+H$. Our model predicts the returns of all the assets $\hat{R}_{t+H}^1, ..., \hat{R}_{t+H}^N$. If we want to maximize our expected profit, we could buy one unit of the asset with the maximum predicted return. However, this strategy is hazardous as our portfolio is not diversified and has a high variance. Therefore, we want to include risk in our objective function. This paper will focus on optimizing the Sharpe ratio (Best, 2010).

Let $w_1, ..., w_N$ be the weights we want to compute for all the assets at time $t$. Using only data from our context, we need to estimate the expected return and volatility to calculate the local Sharpe ratio. The expected return is obtained using the predicted returns: $\sum_{i=1}^N w_i * \hat{R}_{t+H}^i$. For the volatility (i.e., the standard deviation), we have to estimate it based on the assets in the portfolio and the previous values. More precisely, we use the values in the context given to $M_H$. This standard deviation is derived from the covariances of the assets and is given by the formula: $\sqrt{\sum_i \sum_j w_i * w_j * \sigma_{ij}}$, where $\sigma_{ij}$ is the covariance of $X_i$ and $X_j$ during the context.

In the end, when we have to choose the weights, we solve the following problem using traditional optimization techniques (sequential least squares programming in our experiments):

$$\begin{cases} \text{maximize} & \frac{\sum_{i=1}^N w_i * \hat{R}_{t+H}^i}{\sqrt{\sum_i \sum_j w_i * w_j * \sigma_{ij}}} \\ \text{subject to} & \sum_{i=1}^N w_i = 1 \\ & \forall i, 0 \leq w_i \leq 1 \end{cases} \tag{1}$$

Note that we must normalize the return by the size of the horizon H.

### 4.6 FINAL EVALUATION PIPELINE

Finally, this entire evaluation pipeline is the following:

1. Data preprocessing: We clean the data (remove missing values and outliers) and cut large time series into smaller ones that can be taken as input by the models. We also format the data correctly (see Section 5.2).

2. Data split: We split the data chronologically to create a train, validation, and testing dataset (the proportions are 80%, 10%, 10%).

3. Model training: We train each model. The parameters and output might be different between each model.

4. Forecasting evaluation: We use the standard forecasting metrics for the evaluation presented in Section 4.3.

5. Portfolio evaluation: We follow our newly introduced method in Section 4.4.

### 4.7 FINETUNING

During the pretraining of our models, we used our new large dataset. Our goal now is to check if what was learned can be transferred to other smaller datasets. To do so, we finetuned our pretrained models (i.e., we continued the training) on the index datasets S&P500 and CAC40 (Table 1). For the evaluation, we use the same pipeline as previously.

Our approach is inspired by what is done in the natural language processing community: They learn a large model in an unsupervised way (e.g. BERT (Devlin et al., 2018), GPT (Radford et al., 2019)) and reuse it for a specialized problem (e.g., sentiment classification (Gao et al., 2019), information extraction (Kolluru et al., 2020)). Likewise, we pretrained general models on a diversified market and specialized them on many stocks (500) from the US economy (S&P500) and a few stocks (40) from the French economy (CAC40).

## 5 EXPERIMENT SETUP

### 5.1 IMPLEMENTATION

We implemented the algorithms using Python, PyTorch (Paszke et al., 2019), and the code provided by the original authors for the baselines. Our models were trained on machines with two Intel Xeon at 2.40 GHz, 128 GB of RAM, and an Nvidia RTX A6000 with 48GiB of GDDR6. The training time was between 5 days for the smallest models and one month for the biggest ones. For the hyperparameters, we followed the ones given in the literature. We made them vary slightly at the beginning of the training to check they were working well, but we could not run an exhaustive hyperparameters search due to the computational cost. In particular, the learning rate was set at $10^{-4}$ during the initial training on our new dataset and then to $10^{-4}$ during the finetuning. We ran five epochs for each baseline and used an early stop strategy for the finetuning. The code and trained models are or will be provided as additional materials.

For Transformers, we used three configurations. FinTrans$_{312}$ has a latent dimension of 312, 12 layers, 24 heads, and 21.7M parameters. FinTrans$_{1248}$ has a latent dimension of 1248, 24 layers, 24 heads, and 284M parameters. FinTrans$_{2048}$ has a latent dimension of 2048, 32 layers, 32 heads, and 831M parameters. In comparison, SciNet has 105k parameters, and DARNN has 242k parameters.

### 5.2 INPUT REPRESENTATION

In the literature, the points of each time series contain only one dimension: the closing price at a given time $t$, $C_t$. It is the last price on the interval $]t-1, t]$. However, financial data generally come with four other dimensions: the open price $O_t$ (the first price on the interval $]t-1, t]$), the low price (the lowest price on the interval $]t-1, t]$), the high price (the highest price on the interval $]t-1, t]$), and the volume (the number of transactions on the interval $]t-1, t]$).

We are only interested in open, low, high, and close price variations, not their absolute value. For the volume (i.e., the number of transactions), as it can vary by several orders from one asset to another,

we take its log value (we add an $\epsilon$ to prevent problems with $V_t = 0$). Therefore, we transform them as follows ($X = O$, $H$, $L$, or $C$):

$$X'_t = \frac{X_t}{C_{t-1}} \quad \text{and} \quad V'_t = log(V_t + \epsilon) \tag{2}$$

## 6 RESULTS

We first pretrain our models with a horizon of one and a context size of 72 (one day) or 360 (one week) on our large dataset and report the forecasting metrics in Table 2. Then, we finetune the models on S&P500 and CAC40 for three horizons for a context of 72 (the horizons are 1 = 5 minutes, 72 = one day, and 360 = one week) and four horizons for a context size of 360 (the horizons are 1, 72, 360, and 720=one month). For a context of 360, we do not have FinTrans$_{2048}$ as the model becomes too big to fit on a single GPU. We report the forecasting metrics in Table 2 and the portfolio metrics in Table 3. The rest of this section goes more profound in the analysis.

### 6.1 FORECASTING EVALUATION RESULTS

| Dataset | Model | Horizon | RMSE↓ | | | | MAPE↓ | | | |
|---|---|---|---|---|---|---|---|---|---|---|
| Ours (72) | LAST | 1 (pretraining) | **0.0055** | | | | **0.2226** | | | |
| Ours (72) | Scinet | 1 (pretraining) | 0.0144 | | | | 0.9055 | | | |
| Ours (72) | DARNN | 1 (pretraining) | 0.0059 | | | | 0.2873 | | | |
| Ours (72) | FinTrans$_{312}$ | 1 (pretraining) | 0.0096 | | | | 0.6346 | | | |
| Ours (72) | FinTrans$_{1248}$ | 1 (pretraining) | 0.0080 | | | | 0.5174 | | | |
| Ours (72) | FinTrans$_{2048}$ | 1 (pretraining) | 0.0073 | | | | 0.4548 | | | |
| Ours (360) | LAST | 1 (pretraining) | **0.0053** | | | | **0.2143** | | | |
| Ours (360) | Scinet | 1 (pretraining) | 0.0211 | | | | 1.4734 | | | |
| Ours (360) | DARNN | 1 (pretraining) | 0.0126 | | | | 0.9577 | | | |
| Ours (360) | FinTrans$_{312}$ | 1 (pretraining) | 0.0097 | | | | 0.6877 | | | |
| Ours (360) | FinTrans$_{1248}$ | 1 (pretraining) | 0.0083 | | | | 0.6039 | | | |
| S&P500 (72) | LAST | 1 \| 72 \| 360 | **0.00155** | 0.02181 | **0.04909** | | **0.10772** | **1.54404** | 3.57943 | |
| S&P500 (72) | Scinet | 1 \| 72 \| 360 | 0.00346 | 0.02216 | 0.04917 | | 0.25590 | 1.57256 | 3.59490 | |
| S&P500 (72) | DARNN | 1 \| 72 \| 360 | 0.00211 | 0.02186 | 0.04910 | | 0.15861 | 1.54804 | 3.58026 | |
| S&P500 (72) | FinTrans$_{312}$ | 1 \| 72 \| 360 | 0.00157 | 0.02180 | 0.04911 | | 0.11135 | 1.55086 | 3.58086 | |
| S&P500 (72) | FinTrans$_{1248}$ | 1 \| 72 \| 360 | 0.00162 | **0.02179** | **0.04909** | | 0.11420 | 1.54960 | 3.58281 | |
| S&P500 (72) | FinTrans$_{2048}$ | 1 \| 72 \| 360 | **0.00155** | 0.02181 | 0.04910 | | 0.10848 | 1.55215 | 3.58421 | |
| S&P500 (360) | LAST | 1 \| 72 \| 360 \| 720 | **0.00167** | **0.02227** | **0.04903** | **0.06625** | **0.11077** | 1.55516 | 3.56412 | 4.93557 |
| S&P500 (360) | Scinet | 1 \| 72 \| 360 \| 720 | 0.00462 | 0.02266 | 0.04919 | 0.06645 | 0.34894 | 1.58616 | 3.58146 | 4.96633 |
| S&P500 (360) | DARNN | 1 \| 72 \| 360 \| 720 | 0.00235 | 0.02240 | 0.04909 | 0.06633 | 0.12642 | 1.56858 | 3.57438 | 4.95704 |
| S&P500 (360) | FinTrans$_{312}$ | 1 \| 72 \| 360 \| 720 | **0.00167** | 0.02235 | 0.04906 | 0.06669 | 0.11092 | 1.56171 | 3.56822 | 4.95191 |
| S&P500 (360) | FinTrans$_{1248}$ | 1 \| 72 \| 360 \| 720 | **0.00167** | 0.02234 | 0.04904 | 0.06671 | 0.11082 | 1.56315 | 3.56861 | 4.95335 |
| CAC40 (72) | LAST | 1 \| 72 \| 360 | 0.00147 | 0.01756 | 0.03547 | | **0.09939** | 1.23242 | **2.61247** | |
| CAC40 (72) | Scinet | 1 \| 72 \| 360 | 0.00351 | 0.01794 | 0.03612 | | 0.26695 | 1.26915 | 2.67088 | |
| CAC40 (72) | DARNN | 1 \| 72 \| 360 | 0.00150 | **0.01755** | **0.03546** | | 0.10340 | **1.23225** | 2.61285 | |
| CAC40 (72) | FinTrans$_{312}$ | 1 \| 72 \| 360 | 0.00150 | 0.01769 | 0.03553 | | 0.10358 | 1.24465 | 2.61880 | |
| CAC40 (72) | FinTrans$_{1248}$ | 1 \| 72 \| 360 | **0.00146** | 0.01777 | 0.03572 | | **0.09939** | 1.25333 | 2.63667 | |
| CAC40 (72) | FinTrans$_{2048}$ | 1 \| 72 \| 360 | 0.00148 | 0.01766 | 0.03549 | | 0.10098 | 1.24182 | 2.61495 | |
| CAC40 (360) | LAST | 1 \| 72 \| 360 \| 720 | **0.00107** | 0.01692 | 0.03345 | 0.04756 | **0.07428** | 1.17633 | 2.47349 | 3.56498 |
| CAC40 (360) | Scinet | 1 \| 72 \| 360 \| 720 | 0.00397 | 0.01779 | 0.03557 | 0.05221 | 0.30894 | 1.24968 | 2.66036 | 3.99164 |
| CAC40 (360) | DARNN | 1 \| 72 \| 360 \| 720 | 0.00282 | 0.01714 | 0.03340 | 0.04942 | 0.22090 | 1.19866 | 2.46954 | 3.73797 |
| CAC40 (360) | FinTrans$_{312}$ | 1 \| 72 \| 360 \| 720 | 0.00132 | **0.01689** | 0.03323 | 0.04743 | 0.10169 | **1.17434** | **2.45597** | 3.55058 |
| CAC40 (360) | FinTrans$_{1248}$ | 1 \| 72 \| 360 \| 720 | **0.00107** | 0.01719 | **0.03320** | **0.04718** | 0.07513 | 1.20249 | 2.45672 | **3.53021** |

Table 2: Comparison of the baselines for forecasting. Best in bold, second best underlined.

We present the results of our experiments in Table 2. For this table, we reran all the models to ensure comparable results. As we already observed with the code and models from state-of-the-art methods, we can see that the trivial LAST baseline gets the best results on the large-scale dataset. There is an improvement with the finetuning, but no model is statistically significantly better than LAST. This fact confirms that (1) all the models might just be learning noise, and (2) the metrics for time series forecasting (RMSE, MAPE) might not be adapted for financial time series. Here, we do not judge how the models perform on general time series forecasting (electricity, weather, traffic). They generally get better results when clear patterns emerge.

We tested two context sizes on the large dataset: 72 (one day, Ours (72)) and 360 (one week, Ours(360)). Although the scores are similar for the baseline LAST, we can see that all the learned deep models struggle to deal with the additional noise coming from the increased input size from 72 to 360. Only Transformers seems to be able to keep similar performance. In particular, DARNN is the second-best baseline for a context size of 72, but not with a larger context.

Transformers (FinTrans$_{312}$, FinTrans$_{1248}$, FinTrans$_{2048}$) get the second-best results for a context of 360. This is surprising because most approaches claim they beat Transformers. This might be explained by the small amount of data used in previous works that prevented Transformers from being truly effective. In our experiments, we saw that we can increase the size of the Transformers model without overfitting. As we have a large amount of data, we do not need to have a high number of epochs to reach convergence (5 in our case). Some approaches claimed they required over a hundred epochs to get good results.

By looking at the models' predictions, we also observe that the models tend to collapse to the baseline LAST, i.e., consistently predict the last known price or no price variation. This explains why the results for LAST look like a threshold. This collapse improves the performance but makes the network stuck to a local optimum. We observed that the smaller networks, particularly DARNN, were more subject to this problem. This shows these models might be too simple.

The results after the finetuning are similar, although the gap between LAST and the other models narrowed. That leads some models to overperform LAST, but rarely significantly. Only the Transformers models for the CAC40 dataset and a context size of 360 show consistently better results, especially when looking at the MAPE.

## 6.2 Portfolio Evaluation Results

| Dataset | Model | Horizon | Expected Return (%) ↑ | Risk ($e^{-3}$) ↓ | Sharpe Ratio ↑ |
|---|---|---|---|---|---|
| S&P500 (72) | LAST | 1 \| 72 \| 360 | -0.368 \| -0.434 \| 0.017 | 0.750 \| 2.31 \| 1.24 | -4.905(± 0.55) \| -1.878(± 0.36) \| 0.133(± 0.16) |
| S&P500 (72) | Scinet | 1 \| 72 \| 360 | -0.424 \| -0.184 \| **0.065** | 0.638 \| 2.04 \| 1.08 | -6.648(± 0.73) \| **-0.904**(± 0.26) \| **0.601**(± 0.17) |
| S&P500 (72) | DARNN | 1 \| 72 \| 360 | -8.77$e^{-3}$ \| **-0.167** \| 0.035 | 0.512 \| **1.63** \| 1.09 | -0.171(± 0.15) \| -1.022(± 0.27) \| 0.326(± 0.16) |
| S&P500 (72) | Trans$_{312}$ | 1 \| 72 \| 360 | -4.744 \| -3.821 \| 0.019 | 3.75 \| 17.7 \| 1.35 | -12.657(± 1.37) \| -2.156(± 0.40) \| 0.139(± 0.16) |
| S&P500 (72) | Trans$_{1248}$ | 1 \| 72 \| 360 | **0.424** \| -3.754 \| 0.029 | **0.510** \| 17.8 \| **1.02** | **8.328**(± 0.91) \| -2.106(± 0.39) \| 0.284(± 0.16) |
| S&P500 (72) | Trans$_{2048}$ | 1 \| 72 \| 360 | -0.292 \| -3.885 \| 0.040 | 0.570 \| 18.6 \| 1.04 | -5.122(± 0.57) \| -2.086(± 0.39) \| 0.379(± 0.16) |
| S&P500 (360) | LAST | 1 \| 72 \| 360 \| 720 | 0.478 \| **0.033** \| 5.90$e^{-4}$ \| 9.60$e^{-4}$ | 0.595 \| 1.25 \| 1.27 \| 1.23 | 8.042(± 0.89) \| 0.262(± 0.16) \| 4.64$e^{-3}$(± 0.16) \| 7.83$e^{-3}$(± 0.16) |
| S&P500 (360) | Scinet | 1 \| 72 \| 360 \| 720 | 0.364 \| 0.032 \| 0.011 \| **0.023** | 0.448 \| 1.10 \| 1.12 \| 1.05 | 8.130(± 0.90) \| **0.291**(± 0.16) \| 0.096(± 0.16) \| **0.221**(± 0.16) |
| S&P500 (360) | DARNN | 1 \| 72 \| 360 \| 720 | 0.288 \| -2.29$e^{-3}$ \| -0.024 \| 0.017 | 0.322 \| 0.998 \| **0.971** \| **0.965** | 8.944(± 0.91) \| -0.023(± 0.15) \| -0.248(± 0.16) \| 0.174(± 0.16) |
| S&P500 (360) | Trans$_{312}$ | 1 \| 72 \| 360 \| 720 | 0.156 \| 0.027 \| 3.72$e^{-3}$ \| -0.060 | **0.273** \| **0.985** \| 1.10 \| 3.69 | 5.714(± 0.64) \| 0.269(± 0.16) \| 0.034(± 0.16) \| -0.163(± 0.16) |
| S&P500 (360) | Trans$_{1248}$ | 1 \| 72 \| 360 \| 720 | **2.216** \| -0.824 \| **0.042** \| -0.042 | 2.10 \| 6.58 \| 1.13 \| 3.74 | **10.574**(± 1.17) \| -1.252(± 0.21) \| **0.374**(± 0.16) \| -0.113(± 0.16) |
| CAC40 (72) | LAST | 1 \| 72 \| 360 | 1.421 \| 0.128 \| 0.131 | 0.782 \| 1.06 \| 1.00 | 18.159(± 1.97) \| 1.209(± 0.20) \| 1.311(± 0.21) |
| CAC40 (72) | Scinet | 1 \| 72 \| 360 | 0.755 \| 0.128 \| 0.126 | 0.929 \| 1.05 \| 1.44 | 8.127(± 0.89) \| 1.221(± 0.20) \| 0.874(± 0.18) |
| CAC40 (72) | DARNN | 1 \| 72 \| 360 | 1.258 \| **0.155** \| 0.119 | 0.594 \| **0.889** \| **0.984** | **21.170**(± 2.30) \| **1.749**(± 0.24) \| 1.206(± 0.20) |
| CAC40 (72) | Trans$_{312}$ | 1 \| 72 \| 360 | **3.521** \| -0.111 \| 0.143 | 1.76 \| 2.30 \| 1.03 | 19.993(± 2.16) \| -4.84$e^{-1}$(± 0.16) \| 1.393(± 0.22) |
| CAC40 (72) | Trans$_{1248}$ | 1 \| 72 \| 360 | 0.821 \| -0.138 \| **0.178** | **0.460** \| 2.29 \| 1.40 | 17.838(± 1.93) \| -6.02$e^{-1}$(± 0.17) \| 1.269(± 0.21) |
| CAC40 (72) | Trans$_{2048}$ | 1 \| 72 \| 360 | 3.177 \| -0.126 \| 0.131 | 1.75 \| 2.21 \| 1.00 | 18.170(± 1.97) \| -5.72$e^{-1}$(± 0.17) \| 1.309(± 0.21) |
| CAC40 (360) | LAST | 1 \| 72 \| 360 \| 720 | -0.065 \| 0.139 \| 0.129 \| 0.104 | 0.289 \| 0.998 \| 0.930 \| 0.970 | -2.259(± 0.29) \| 1.394(± 0.22) \| 1.386(± 0.22) \| 1.069(± 0.20) |
| CAC40 (360) | Scinet | 1 \| 72 \| 360 \| 720 | -0.547 \| 0.282 \| **0.224** \| **0.154** | 0.336 \| 1.18 \| 1.67 \| 1.70 | -16.268(± 1.78) \| **2.385**(± 0.30) \| 1.338(± 0.21) \| 0.906(± 0.19) |
| CAC40 (360) | DARNN | 1 \| 72 \| 360 \| 720 | -0.248 \| 0.184 \| 0.143 \| 0.139 | 0.346 \| 0.916 \| 0.835 \| 1.66 | -7.187(± 0.80) \| 2.006(± 0.27) \| 1.706(± 0.24) \| 0.839(± 0.18) |
| CAC40 (360) | Trans$_{312}$ | 1 \| 72 \| 360 \| 720 | -0.267 \| 0.090 \| 0.126 \| 0.086 | **0.256** \| **0.804** \| 0.735 \| 1.03 | -10.437(± 1.15) \| 1.124(± 0.20) \| **1.712**(± 0.24) \| 0.843(± 0.18) |
| CAC40 (360) | Trans$_{1248}$ | 1 \| 72 \| 360 \| 720 | **0.313** \| **0.512** \| 0.120 \| 0.122 | 1.17 \| 2.31 \| **0.725** \| **0.878** | **2.669**(± 0.33) \| 2.222(± 0.29) \| 1.648(± 0.24) \| **1.394**(± 0.22) |

Table 3: Comparison of the baselines for portfolio metrics. Best in bold. Confidence interval at 95% for Sharpe Ratio.

We present the results of the portfolio evaluation in Table 3. We first notice that even though the RMSE and MAPE metrics were close after finetuning, the portfolio metrics show larger and more significant gaps. It proves we are now looking at the problem from a new angle, closer to real-life scenarios. Besides, the new metrics are uncorrelated with the forecasting metrics. In particular, LAST is very rarely the best model, showing that we trained models that can make better decisions.

However, we observe a high variance amongst the results, amplified on short horizons. In addition, the results do not allow us to conclude whether one architecture is better. These two observations have the same cause: all the models were trained to optimize an objective function similar to RMSE. Therefore, they did not learn to make good and consistent decisions. Indeed, RMSE measures the difference between the gold standard and the prediction and does not care if the prediction is above or below the truth. Nevertheless, this information is crucial when deciding to buy or sell.

Our results show the limitations of the current models when dealing with financial time series forecasting. By accepting the complexity and particularities of this problem, we can fully leverage the power of deep models. This new research direction must rely on new metrics based on decision-making and not absolute predictions. This is true for both the evaluation (as we show in this paper) and the training objective.

### 6.3 LIMITATIONS

We reused similar parameters for the baselines as those provided in the original papers. We could have finetuned them better and increased the size of the networks, but it would have taken much more time because of the size of the dataset. Besides, the original authors ran an exhaustive search on hyperparameters. We only trained large networks for Transformers as they have a well-known architecture and can later be reused for tasks with different context sizes.

We focused on evaluating models trained only for the forecasting task, not for the ranking task involved in portfolio construction. The paper explores more adapted financial time series forecasting metrics, but specialized non-forecasting models might get better results. We also compared deep learning models and did not consider simpler models. We decided to concentrate on the active domain of deep learning and showed in this paper that simpler models are hard to beat, depending on what we measure. Yet, deeper models are more scalable and can improve results if evaluated correctly.

For the portfolio metrics, we ignored some parameters that usually reduce the performances but have a limited impact on the relative comparison between the baselines. In particular, we did not consider that one must pay a fee to make a trade, and we ignore the risk-free rate in the Sharpe ratio. This last parameter allows traders to evaluate a strategy relative to a "risk-free" strategy.

Finally, the testing period of our models takes place in a historical moment when the financial market faced post-Covid, Ukrainian war, and high inflation. These problems make the task of forecasting and portfolio building even harder. The fact we mostly have positive Sharpe ratios is encouraging.

### 7 CONCLUSION

In this paper, we study the task of financial time series forecasting. We saw that the current deep learning models have a blind spot: They often compare their results with each other, but they fail to beat the trivial baseline that simply predicts the last known point. This fact was observable in their limited datasets and the large-scale dataset we used in this article. From this observation, we noted that our trivial baseline is, in fact, very bad at making decisions, which is crucial in finance when constructing portfolios. Therefore, we proposed to use a new evaluation pipeline based on portfolio construction: We mapped the predictions of each model to an investment strategy, and we evaluated this strategy.

From this new set of experiments, we observed that deep models are better at making decisions in some cases, but as they were not trained for this task, they have a high variance in their results. Therefore, we suggested rethinking how we train models for financial time series forecasting: We need to remember that the final goal is to make a decision. That must impact the training objectives and evaluation metrics.

In this paper and many state-of-the-art papers, the forecasting solely relies on the values from the time series. However, the price of assets is strongly impacted by external events (financial reports, war, elections). The difficulties we observed might also indicate that we must include more semantic information about the companies or the news in the predictions.

Although we cannot make our dataset public for legal reasons, we provide the code and the pre-trained models used in our experiments. We hope that future works will be more critical regarding financial time series forecasting, will adapt their evaluation framework, and will bootstrap their models with our pre-trained models. This way, they should be able to advance the state-of-the-art, even when working with limited datasets.

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
