# OpenReview forum: "Evaluating and Finetuning Models For Financial Time Series Forecasting"
_ICLR.cc/2024/Conference — Submitted to ICLR 2024_

### Official Review · Reviewer_BmfC · 2023-10-30

**Soundness:** 2 fair
**Presentation:** 3 good
**Contribution:** 2 fair
**Rating:** 5
**Confidence:** 4

**Summary:**

This paper proposed to evaluate the financial time series forecasting models by deciding which assets to buy and sell, i.e., form a portfolio, according to the prediction, rather than comparing the errors between predictions and ground-truths. The authors collected a dataset from S&P500 and CAC40, and evaluated several forecasting models (e.g., Scinet and DARNN) using the proposed evaluation pipeline.

**Strengths:**

* In general, the idea of indirect comparison, through the proxy of portfolio optimisation and backtesting, is an interesting alternative to the direct comparison, i.e., on prediction errors.

* The optimisation over Sharpe ratio is reasonable choice under the context.

* The authors found that "LAST", the very simple baseline of always predicting the next value as its current one, is surprisingly hard to beat, which was often ignored in previous works.

**Weaknesses:**

* It remains unclear how the datasets were collected. The authors claimed that "we cleverly used web scrapping tools" and the data sources are unknown thus the quality of data might be questionable.

* The transaction cost is not discussed in this paper, so the metrics of portfolio may not be meaningful as the impact of transaction cost is significant, given the reasonably frequent rebalance.

**Questions:**

The objective function in (1) does not look like a trivial problem, esp. given the non-negative and sum-to-one constraints, what was the optimiser used in this paper? did it converge?

---

> ### Author Response · Authors · 2023-11-21
>
> General Note:
>
> We thank the reviewers for their valuable feedback. We hope to address their concerns in this rebuttal.
>
> Before answering each concern, we want to recall our contribution and why it is essential.
>
> It is well known that constructing good models is hard for finance professionals. It is tough to beat simple baselines. When we studied how deep learning forecasting methods can be applied to our problem, we were surprised to notice that many papers have been published on the topic, all claiming to beat previous state-of-the-art. These papers were published in top conferences and had a significant impact on the community (SciNet, NeurIPS 2022, 66 citations, DARNN, IJCAI 2017, 1200 citations, N-BEAT, ICLR 2020, 750 citations, N-HITS, AAAI 2023, 100 citations, ...). When we tried to reproduce the results, we could not beat trivial random baselines, which was very concerning. Therefore, we decided to compare financial forecasting models in a unified way (on the same dataset) and still observed worse than random performance. We tested two hypotheses that might break this barrier.
>
> 1) We increased the training dataset size (usually, the more data, the better). We will provide a script to reconstruct our dataset and reproduce all our results.
>
> 2) We increased the model size (which should be possible thanks to the first point). Because training is costly, we only tested the impact of size on the Transformers architecture.
>
> Both these hypotheses were wrong, so the problem was more profound. Our investigations led us to a flaw in the evaluation process, and we proposed a new approach to evaluate financial forecasting models. With this new pipeline, we had encouraging results that will lead to thrilling future works.
>
> We think our contribution is worthwhile. It seems obvious now that we exposed the problem and connected the dots. However, previous papers published at top conferences (including ICLR, NeurIPS, and AAAI), reviewed by the community and cited by many works, did not correctly evaluate their results. Nobody noticed before. Publishing complex models without a good evaluation seems useless to us.
>
> If the reviewers still want to reject our paper after this rebuttal, we would like their input on the following question: Is it worth spending time to better understand already published models and why they perform worse than random in certain cases? If yes, what is missing in our paper to change how financial forecasting models are evaluated?

---

> > ### Author Response · Authors · 2023-11-21
> >
> > * About the dataset, we do not believe the technical details are interesting for the paper. We reverse-engineered several public APIs and combined the results. We can add them to a technical report. However, please note that the code to gather the data will be provided, thus making it possible for anyone to reconstruct the dataset.
> > * We noted in the limitations that the transaction costs are ignored, as all models will pay the same amount of money as they all make the same number of transactions. Therefore, we explicitly said in the limitations that the numbers are overestimated but remain relatively comparable among themselves.
> > * As the reviewer noticed, the optimization problem is non-trivial, making it impossible to use linear optimization. As mentioned in the text, we used sequential least squares programming in our experiments and observed no convergence problems.

---

### Official Review · Reviewer_Zb2D · 2023-11-01

**Soundness:** 3 good
**Presentation:** 3 good
**Contribution:** 1 poor
**Rating:** 5
**Confidence:** 4

**Summary:**

This article points out the existing imperfection of the evaluation pipeline used for financial time series forecasting. From this observation, the article explores the specific reason hidden in this phenomenon and put forward a brand-new evaluation pipeline based on portfolio construction, whose main idea lies at mapping the predictions of each model to an investment strategy. Based on this new evaluation pipeline, the authors' team test many baseline models' effect in financial time series forecasting task.
The contributions of this article contains:
1. This article introduce a new evaluation pipeline better suited for financial time series.
2. This article compare state-of-the-art deep learning methods for financial time series forecasting based on the brand-new evaluation pipeline.

**Strengths:**

1. originality: This article is innovative, figuring out the existing problem in evaluation pipeline of financial time series forecasting，and putting forward a brand-new angle to evaluate the effect.
2. quality: The logical chain of this article is relatively complete, from definition of financial time series forecasting problem, the methodology of experimenting, to the final result of their experiments and their conclusions.
3. clarity: The article is relatively clear，but the meaning of many variables involved in formulas are not mentioned, which confused me a lot.
4. significance: The contributions of this article are not significant. On the one hand, the evaluation pipeline put forward in this article only works for financial time series forecasting, which is not general. On the other hand, this article claim they are training a general model which could be used for many specific tasks, but they didn't list any experiment result about this "general model".

**Weaknesses:**

1. The meaning of many variables involved in formulas are not mentioned. For example, the $P_t$ put forward in section 3, which should be relevant with time step variant $t$, but the formula does not contain $t$.
2. This passage is aimed to address the existing problem in the evaluation pipeline of financial time series forecasting, which means one of this passage's key point is putting forward the existing problem, but the problem is mentioned in a paragraph in section 6.1 " FORECASTING EVALUATION RESULTS". It's too "convert" to figure out, making it difficult for me to grasp the logical lines of the entire article.
3. The author mentioned that one of the contribution of this article is "We train large models on a large dataset"，and "We show that these models can be used to solve more specific tasks". But he did not list the experiment result of these "general models" trained by his team，and there was no experiment result that could prove their general models can be used to solve more specific tasks.

**Questions:**

1. About the definition of $R^i_j$ which is put forward in section 4.4，the formula to calculate $R^i_j$ seems irrelevant with $j$，what's the meaning of $j$ ？
2. About the definition of $P_t$ (the value of a portfolio at time t) which is put forward in section 3，the formula to calculate $P_t$ seems irrelevant with time step $t$，does the time step $t$ influence $x^i_k$，or $w_i$ ？
3. In the experiment, why not control the scale(the number of parameters) of baseline models？If the  scales of baseline models are not consistent，are the result convincing？

**Details Of Ethics Concerns:**

Not applicable.

---

> ### Author Response · Authors · 2023-11-21
>
> General Note:
>
> We thank the reviewers for their valuable feedback. We hope to address their concerns in this rebuttal.
>
> Before answering each concern, we want to recall our contribution and why it is essential.
>
> It is well known that constructing good models is hard for finance professionals. It is tough to beat simple baselines. When we studied how deep learning forecasting methods can be applied to our problem, we were surprised to notice that many papers have been published on the topic, all claiming to beat previous state-of-the-art. These papers were published in top conferences and had a significant impact on the community (SciNet, NeurIPS 2022, 66 citations, DARNN, IJCAI 2017, 1200 citations, N-BEAT, ICLR 2020, 750 citations, N-HITS, AAAI 2023, 100 citations, ...). When we tried to reproduce the results, we could not beat trivial random baselines, which was very concerning. Therefore, we decided to compare financial forecasting models in a unified way (on the same dataset) and still observed worse than random performance. We tested two hypotheses that might break this barrier.
>
> 1) We increased the training dataset size (usually, the more data, the better). We will provide a script to reconstruct our dataset and reproduce all our results.
>
> 2) We increased the model size (which should be possible thanks to the first point). Because training is costly, we only tested the impact of size on the Transformers architecture.
>
> Both these hypotheses were wrong, so the problem was more profound. Our investigations led us to a flaw in the evaluation process, and we proposed a new approach to evaluate financial forecasting models. With this new pipeline, we had encouraging results that will lead to thrilling future works.
>
> We think our contribution is worthwhile. It seems obvious now that we exposed the problem and connected the dots. However, previous papers published at top conferences (including ICLR, NeurIPS, and AAAI), reviewed by the community and cited by many works, did not correctly evaluate their results. Nobody noticed before. Publishing complex models without a good evaluation seems useless to us.
>
> If the reviewers still want to reject our paper after this rebuttal, we would like their input on the following question: Is it worth spending time to better understand already published models and why they perform worse than random in certain cases? If yes, what is missing in our paper to change how financial forecasting models are evaluated?

---

> > ### Author Response · Authors · 2023-11-21
> >
> > * We corrected the mistakes on the indices.
> > * "This passage". Can the reviewer explicitly say what passage they are talking about? We will make the problem clearer earlier in the text if this is the problem.
> > * About the finetuning, the general models were finetuned on more specific time series, which are the assets in S&P500 and CAC40. The results are in Tables 2 and 3. This is what we call more specific tasks: a specific market. However, we believe our general models can be applied to other tasks, such as price direction prediction (with a classifier).
> > * Question 3: We studied the impact of the size with a simpler model: Transformers. This is because the training is costly, and the original authors of the baselines claimed that their models beat state-of-the-art results with these parameters. We believe all models will behave more or less similarly to Transformers, although we could try with a wider variety of models (which would be more expensive).

---

### Official Review · Reviewer_9z9f · 2023-11-03

**Soundness:** 3 good
**Presentation:** 2 fair
**Contribution:** 1 poor
**Rating:** 5
**Confidence:** 3

**Summary:**

In this work, the authors conduct a benchmarking study on deep learning applied to financial time series forecasting.


The authors construct a dataset of financial time series, including the price and volume histories of various stocks, options, etc.
On this dataset, the authors pre-train several relevant deep learning (DL) approaches. These models are then fine-tuned to forecast the price and volume of stocks in the S&P500 and CAC40 indices. The authors demonstrate that a naïve baseline outperforms DL approaches on their selected tasks when evaluated using standard forecasting metrics. To address this, they propose an evaluating performance based on the returns and risks associated with using the DL approaches for portfolio management. They find that DL tends to perform better than the baseline under this evaluation.

**Strengths:**

- The experimental evaluation is thorough, although the authors are encouraged to more completely describe their methodology.
- The authors provide convincing empirical evidence that DL methods struggle to beat baseline methods.

**Weaknesses:**

- Although the evaluation of Scinet and DA-RNN on finance data is novel as far as I am aware, the benchmarking of DL on financial data is not novel. For example, [1] evaluates Transformers on several indexes, and also considers the risk/return on trading strategies based on DL forecasts. As the authors point out, their benchmarking submission also does not include comparison to other families of time series forecasting methods.

- While the authors claim to collect a comprehensive dataset, no description is given other than the number of samples. This makes evaluation of the significance and quality of their dataset difficult. Furthermore, while a complete collection of financial price histories could be beneficial to the community, it is ultimately not difficult to assemble publicly available historical price data, making this contribution limited.


- I disagree with the author's characterization that the naïve or LAST baseline (predicting the last known point) has been ignored in forecasting of financial time series. See [2], where a widely used textbook states that the naïve approach works well in financial data. Furthermore, the authors should consider the existence of metrics such as the Mean Absolute Scaled Error (MASE), which compares forecast performance against the naïve one-step forecast model. The M3/M4 papers as cited in this submission also perform evaluation against the naïve baseline. While individual papers in the financial forecasting literature (such as [1]) do fail to make adequate comparison, I believe the authors claims that this is a systemic issue needs further support.


- The authors are encouraged to add more clarity in the writing of this submission. While the flow and core narrative of this submission is clear, it often does not contain enough detail for critical evaluation. Also, table 3 appears to exceed the allowable margins.


[1] Wang, Chaojie, et al. "Stock market index prediction using deep Transformer model." Expert Systems with Applications 208 (2022): 118128.

[2] https://otexts.com/fpp2/simple-methods.html#na%C3%AFve-method

**Questions:**

- The submission would benefit from a brief survey of any existing approaches which evaluate forecasting models based on portfolio performance. Given that the expected return, risk and Sharpe ratio are widely used metrics, it would be beneficial to understand the novelty of the portfolio evaluation proposed in this submission.

- Any rebuttal to the above weaknesses would be appreciated.

---

> ### Author Response · Authors · 2023-11-21
>
> General Note:
>
> We thank the reviewers for their valuable feedback. We hope to address their concerns in this rebuttal.
>
> Before answering each concern, we want to recall our contribution and why it is essential.
>
> It is well known that constructing good models is hard for finance professionals. It is tough to beat simple baselines. When we studied how deep learning forecasting methods can be applied to our problem, we were surprised to notice that many papers have been published on the topic, all claiming to beat previous state-of-the-art. These papers were published in top conferences and had a significant impact on the community (SciNet, NeurIPS 2022, 66 citations, DARNN, IJCAI 2017, 1200 citations, N-BEAT, ICLR 2020, 750 citations, N-HITS, AAAI 2023, 100 citations, ...). When we tried to reproduce the results, we could not beat trivial random baselines, which was very concerning. Therefore, we decided to compare financial forecasting models in a unified way (on the same dataset) and still observed worse than random performance. We tested two hypotheses that might break this barrier.
>
> 1) We increased the training dataset size (usually, the more data, the better). We will provide a script to reconstruct our dataset and reproduce all our results.
>
> 2) We increased the model size (which should be possible thanks to the first point). Because training is costly, we only tested the impact of size on the Transformers architecture.
>
> Both these hypotheses were wrong, so the problem was more profound. Our investigations led us to a flaw in the evaluation process, and we proposed a new approach to evaluate financial forecasting models. With this new pipeline, we had encouraging results that will lead to thrilling future works.
>
> We think our contribution is worthwhile. It seems obvious now that we exposed the problem and connected the dots. However, previous papers published at top conferences (including ICLR, NeurIPS, and AAAI), reviewed by the community and cited by many works, did not correctly evaluate their results. Nobody noticed before. Publishing complex models without a good evaluation seems useless to us.
>
> If the reviewers still want to reject our paper after this rebuttal, we would like their input on the following question: Is it worth spending time to better understand already published models and why they perform worse than random in certain cases? If yes, what is missing in our paper to change how financial forecasting models are evaluated?

---

> > ### Author Response · Authors · 2023-11-21
> >
> > About [1], we believe it is a work of poor quality:
> > * The datasets used contain four times series of 2,500 data points each (daily price for ten years, 2000 for training, 500 for testing). This is not enough to train a deep model. Therefore, the observed results are very likely to be noise.
> > * The paper fails to notice that their models are worse than random on standard metrics.
> > * Although they use metrics similar to ours (return, volatility, Sharpe ratio, standard metrics in finance), they first transform their forecasting model into a classifier. Therefore, they badly evaluate the forecasting performance of the model.
> > * They used a classifier representation because there was only a one-time series at the time. In real life, there are several simultaneously, and an investor has to choose in which proportions each asset is bought. We proposed this in our paper, and our approach better evaluates the forecasting capabilities.
> > * The results are not reproducible. In particular, the B&H baseline has very poor results. On the S&P 500, they report a Sharpe ratio of 0.53, whereas it was between 1 and 2 over the testing period.
> > * For some reason, the return contains a log function.
> > * As noted by one of the reviewers and as we do in the paper, it is better to predict a given stock's return directly rather than an absolute price.
> >
> > About the previous work on evaluating forecasting metrics using portfolio metrics, we will check again the earlier work, but we previously found no serious work on this topic.
> >
> > About the dataset, we do not believe the technical details are interesting for the paper. We reverse-engineered several public APIs and combined the results. We can add them to a technical report. However, please note that the code to gather the data will be provided, thus making it possible for anyone to reconstruct the dataset.
> >
> > We are conscious that the naive baseline is something well-known in finance. We were surprised when we worked on the previous works on deep learning as they always claim to improve over previous models but never return to the trivial baseline. Most deep learning papers from people outside the finance realm ignore this baseline. These papers are published in top conferences and have many citations, which shows the problem is quite serious and ignored by our community (SciNet, NeurIPS 2022, 66 citations, DARNN, IJCAI 2017, 1200 citations, N-BEAT, ICLR 2020, 750 citations, N-HITS, AAAI 2023, 100 citations, ...).

---

> ### Comment · Reviewer_9z9f · 2023-11-21
>
> I would like to first thank the authors for their rebuttal.
>
> Overall, the author's point on the consistency of forecasting evaluation is valid. However, solely based on content in this submission, it is hard to understand whether this is a problem with individual papers, or a systemic issue in the community. Though individual papers fail to make comparisons against good baselines, this is not the case for all papers. For example, while papers such as N-BEATS or N-HITS don't make comparison against LAST, they do compare against other simple statistical baselines. As previously pointed out, the M3/M4 papers, while not specifically investigating finance, do make exact comparisons to the LAST baseline. I believe a more comprehensive related works section and expanding the experimental section to compare against N-BEATS / N-HITS and other papers could better support the position of this submission.
>
> Ultimately, while I have increased my score due to the rebuttal, I believe this paper makes a large (though potentially true) claim that is not fully support in the manuscript due to a lack of in-depth textual and experimental comparison against existing works and their shortcomings.
>
> Specific points:
> - While the technical details may not be "interesting" for the main text, a description of the process would have allowed the reviewers to independently evaluate one of the claimed key contributions of this submission. Thank you for offering to add this to the technical report or code in the supplementary, I believe this will strengthen the manuscript.
>
> - Thank you for outlining the issues with the example citation. As above, I believe the comparison of this submission against others papers that apply DL to finance should be expanded in the related works section. This would better convince readers of the shortcomings of existing literature.

---

### Official Review · Reviewer_pP2m · 2023-11-07

**Soundness:** 2 fair
**Presentation:** 2 fair
**Contribution:** 1 poor
**Rating:** 3
**Confidence:** 3

**Summary:**

The authors propose model construction pipeline to be used to benchmark methods for returns forecasting and portfolio construction in financial applications. The author also proposes a procedure to transfer pre-trained models onto smaller datasets and fine tuned for trading applications.

**Strengths:**

Given the diversity of time series datasets, having access to a high quality specialist dataset for finance can be useful. Knowledge of standardised approaches to evaluate and benchmark both forecasting and portfolio construction methods can be useful for practitioners.

**Weaknesses:**

While the paper does put together a sequence of standardised techniques (for evaluation and for transfer learning), it fails to 1) concretely demonstrate what novel methods have been proposed and 2) why existing methods are insufficient.

On the evaluation front, numerous papers have been proposed to evaluate machine learning-based trading strategies (see references for both forecasting and portfolio construction), including 1) which benchmarks a variety of techniques in a standardised fashion. All of which have not been referenced by authors.

Furthermore, in contrast to claims that standardised datasets are lacking -- numerous open-source financial datasets can be found, and a list has been supplied below. The authors themselves do not open source their dataset (citing legal reasons that prevent publication), which run slightly contrary to the goal of developing a common framework for benchmarking.

In addition, transfer learning in the financial domain is also not a novel idea, and comparisons to previous works are absolutely required.

References
--
1. Gu et al 2020. Empirical Asset Pricing via Machine Learning, The Review of Financial Studies, Volume 33, Issue 5
2. Poh et al 2021. Building Cross-Sectional Systematic Strategies by Learning to Rank. The Journal of Financial Data Science Spring 2021.
3. Marcos Lopez de Prado 2016. Building Diversified Portfolios that Outperform Out-of-sample. Journal of Portfolio Management.
2. Koshiyama et al 2022. QuantNet: transferring learning across trading strategies, Quantitative Finance, 22:6, 1071-1090


Kaggle competitions/ datasets
---
1. M6 competition -- https://github.com/Mcompetitions/M6-methods
2. G-Research Crypto Forecasting -- https://www.kaggle.com/competitions/g-research-crypto-forecasting
3. JPX Tokyo Stock exchange Prediction -- https://www.kaggle.com/competitions/jpx-tokyo-stock-exchange-prediction
4. Kaggle sample NASDAQ dataset -- https://www.kaggle.com/datasets/jacksoncrow/stock-market-dataset

**Questions:**

1. Why is evaluation only performed on one day or one week of data? Most finanical papers test strategies over multiple years.
2. Why is the proposed pipeline superior to existing methods for evaluating trading strategies?
3. Is MSE benchmarked with regards to price forecasts (as seen from LAST)? Would returns or price change forecasts (with naive benchmark being returns=0) be more suitable approach given the non-stationarity of price data?

---

> ### Author Response · Authors · 2023-11-21
>
> General Note:
>
> We thank the reviewers for their valuable feedback. We hope to address their concerns in this rebuttal.
>
> Before answering each concern, we want to recall our contribution and why it is essential.
>
> It is well known that constructing good models is hard for finance professionals. It is tough to beat simple baselines. When we studied how deep learning forecasting methods can be applied to our problem, we were surprised to notice that many papers have been published on the topic, all claiming to beat previous state-of-the-art. These papers were published in top conferences and had a significant impact on the community (SciNet, NeurIPS 2022, 66 citations, DARNN, IJCAI 2017, 1200 citations, N-BEAT, ICLR 2020, 750 citations, N-HITS, AAAI 2023, 100 citations, ...). When we tried to reproduce the results, we could not beat trivial random baselines, which was very concerning. Therefore, we decided to compare financial forecasting models in a unified way (on the same dataset) and still observed worse than random performance. We tested two hypotheses that might break this barrier.
> 1) We increased the training dataset size (usually, the more data, the better). We will provide a script to reconstruct our dataset and reproduce all our results.
> 2) We increased the model size (which should be possible thanks to the first point). Because training is costly, we only tested the impact of size on the Transformers architecture.
> Both these hypotheses were wrong, so the problem was more profound. Our investigations led us to a flaw in the evaluation process, and we proposed a new approach to evaluate financial forecasting models. With this new pipeline, we had encouraging results that will lead to thrilling future works.
>
> We think our contribution is worthwhile. It seems obvious now that we exposed the problem and connected the dots. However, previous papers published at top conferences (including ICLR, NeurIPS, and AAAI), reviewed by the community and cited by many works, did not correctly evaluate their results. Nobody noticed before. Publishing complex models without a good evaluation seems useless to us.
>
> If the reviewers still want to reject our paper after this rebuttal, we would like their input on the following question: Is it worth spending time to better understand already published models and why they perform worse than random in certain cases? If yes, what is missing in our paper to change how financial forecasting models are evaluated?

---

> ### Author Response · Authors · 2023-11-21
>
> Rebuttal:
>
> *Question 1*: The context size is one day or one week, but the training period spreads over several years. The models limit the context size. We can have such a context because we have access to data at a low granularity. Almost all previous works have a granularity of one day, which increases the time-span of data in the context. However, as noted in the introduction, training data can quickly become obsolete. Future work remains on integrating broader time periods into low-granularity models.
>
> *Question 2*: Our goal is not to evaluate trading strategies. This is an old problem, and we did not claim to have invented anything on this topic (return, volatility, and Sharpe Ratio are standrd metrics). What we proposed in our paper is to offer a methodology to evaluate financial time series forecasting models.
>
> *Question 3*: We do not predict a price but a return (see 5.2). The transition between the representations can quickly be done.
>
> About the datasets cited by the reviewer, they are samples of a specific market. We never claimed these samples did not exist, but there was a lack of an extensive dataset covering more instruments, assets, markets, and granularity :
> * M6: This repository does not contain data, but, like us, a script to gather the data. The data includes 100 time series at a daily granularity. Even if the data ranges over 20 years, we get 500k data points, which is 1000 times less than us. Besides, we have a wider variety of instruments and a finer granularity (minute instead of day).
> * G-Research Crypto Forecasting: This dataset only contains crypto (we have them too). It contains 13 time series and 24 million data points at the minute granularity. This is 42 times less than us. Besides, we have a wider variety of instruments, and data about cryptocurrencies are easy to gather as they are public. Cryptocurrencies also behave differently than standard assets and FOREX.
> * JPX Tokyo Stock Exchange Prediction: This dataset contains daily prices of 4.4k Japanese companies for a total number of data points of 2.3 million. In comparison, we have 500 times more data, a wider variety of instruments, and a finer granularity (minute instead of day).
> * NASDAQ 100: This dataset is already mentioned in the paper (see Table 1)
>
> The references given by the authors are very interesting. We will include a paragraph on non-forecasting model evaluations. In particular, [1] suggests evaluating forecasting models by building a portfolio. However, we found key differences in our approach. First, the motivation is different: They found no problems with other metrics (because they use simple models only). Second, the evaluating setting is different. Their models do not take as input time series but a few (94) manually designed features that include exogenous information (not derived from the time series alone). Each month, for 30 years, they construct a portfolio. Therefore, the testing set contains only 360 data points. Besides, the training and testing spread over 60 years. It was shown (see reference in our paper) that models have trouble adapting over such long periods of time because of changes in legislation or trading techniques. Third, the forecasting output is only used to rank the assets into deciles. In the end, a maximum of 20% of the assets are kept (and therefore, the model is not really evaluated on 80% of the assets). An external source already gives the weights of the portfolio. The forecasting model is just here to filter the assets. In our case, we propose using all the predictions from the forecasting model (and nothing else) and constructing a portfolio to maximize the Sharpe Ratio (both keeping a high return and low volatility). Therefore, we better evaluate the performance of the forecast without any external help. This approach is novel. However, we will cite this suggested paper and discuss our article's differences.
>
> For the other papers, [2] works like [1]. [3] applies machine learning techniques (not models) to a mathematical problem but is not related to our problem. [4] is not a forecasting model; it directly predicts the weights in the portfolio, knowing the entire market. This is not the case in our paper and the previous work papers we compared. The reported Sharpe ratio scores of the baselines are very low compared to the S&P500 at the time, which is strange.

---

### Meta-Review · Area_Chair_fKvi · 2023-12-11

**Metareview:**

This paper creates a pipeline to benchmark methods for forecasting returns and building portfolios in financial applications. This was motivated by the desire to have a realistic benchmark for ML models in finance and because a simple baseline of forward fill is never evaluated in prior work (despite it being difficult to beat). The reviewers found the work interesting and useful but had a difficult time evaluating how comprehensive the proposed dataset is since no information was provided on it other than the number of samples. The authors response to this question is "we do not believe the technical details are interesting for the paper".

While I think this to be an important problem and commend the authors for putting together this research effort, I do not think it is feasible for reviewers to evaluate the utility of the benchmark based on such little information since the benchmark is one of the claimed contributions of this work. For this work to have an impact in the community the data resource itself and details on its creation are as important as the verification of the claim that the proposed benchmark is useful.

In its current form, while the experiments are detailed and the problem identified useful for the community to know about the manuscript falls short on two dimensions. First, I think it *is* both relevant and important for the text of the paper contain technical details (supplementary material is fine) on how the data were put together.  I encourage the authors to put together a detailed anonymous code and data link to enable reviewers to verify this in any future submission. The benchmark and datasets track at NeurIPS the last few years provides several such examples of how to format and present such a resource for the community -- I encourage the authors to review the same.Second, as per 9z9f's suggestion, it is worth putting together a comparison of this submission against other DL for finance work within an expanded related works.

**Justification For Why Not Higher Score:**

Justification provided in main review.

**Justification For Why Not Lower Score:**

N/A

---

### Decision · Program_Chairs · 2024-01-16

Reject